# Super-Resolution Imaging of Fas/CD95 Reorganization Induced by Membrane-Bound Fas Ligand Reveals Nanoscale Clustering Upstream of FADD Recruitment

**DOI:** 10.3390/cells11121908

**Published:** 2022-06-12

**Authors:** Nicholas Frazzette, Anthony C. Cruz, Xufeng Wu, John A. Hammer, Jennifer Lippincott-Schwartz, Richard M. Siegel, Prabuddha Sengupta

**Affiliations:** 1Immunoregulation Section, Autoimmunity Branch, National Institute of Arthritis and Musculoskeletal and Skin Diseases (NIAMS), National Institutes of Health, Bethesda, MD 20892, USA; nfrazzette@gmail.com (N.F.); cruza2@mail.nih.gov (A.C.C.); 2Cell and Developmental Biology Center, National Heart, Lung and Blood Institute, National Institutes of Health, Bethesda, MD 20892, USA; wux@nhlbi.nih.gov (X.W.); hammerj@nhlbi.nih.gov (J.A.H.); 3Janelia Research Campus, Howard Hughes Medical Institute, Ashburn, VA 20147, USA; lippincottschwartzj@hhmi.org

**Keywords:** Fas, CD95, PALM imaging, super-resolution microscopy, receptor signaling, TNFR superfamily

## Abstract

Signaling through the TNF-family receptor Fas/CD95 can trigger apoptosis or non-apoptotic cellular responses and is essential for protection from autoimmunity. Receptor clustering has been observed following interaction with Fas ligand (FasL), but the stoichiometry of Fas, particularly when triggered by membrane-bound FasL, the only form of FasL competent at inducing programmed cell death, is not known. Here we used super-resolution microscopy to study the behavior of single molecules of Fas/CD95 on the plasma membrane after interaction of Fas with FasL on planar lipid bilayers. We observed rapid formation of Fas protein superclusters containing more than 20 receptors after interactions with membrane-bound FasL. Fluorescence correlation imaging demonstrated recruitment of FADD dependent on an intact Fas death domain, with lipid raft association playing a secondary role. Flow-cytometric FRET analysis confirmed these results, and also showed that some Fas clustering can occur in the absence of FADD and caspase-8. Point mutations in the Fas death domain associated with autoimmune lymphoproliferative syndrome (ALPS) completely disrupted Fas reorganization and FADD recruitment, confirming structure-based predictions of the critical role that these residues play in Fas–Fas and Fas–FADD interactions. Finally, we showed that induction of apoptosis correlated with the ability to form superclusters and recruit FADD.

## 1. Introduction

The Fas/CD95 receptor, a member of the TNF receptor superfamily, induces apoptosis via the generation of a death-inducing signaling complex (DISC). This complex transduces the association of Fas with its ligand (FasL) into caspase-8 activation through the recruitment of the Fas-associated death domain (FADD) protein. Fas recruits FADD via association between the intracellular Fas receptor death domain and the FADD death domain. FADD, in turn, recruits caspase-8 via association between the FADD death effector domain and the caspase-8 death effector domain [1]. T cell-induced apoptosis via this pathway prevents autoimmunity and accumulation of CD4^−^CD8^−^ T cells in the peripheral immune system, as occurs in the autoimmune lymphoproliferative syndrome associated with dominant-negative mutants of Fas [2,3,4]. Fas signaling can also trigger non-apoptotic functions [5] including T cell costimulation [6], accelerated T cell differentiation that impairs antitumor immunity [7], neuronal differentiation, [6] tumor growth and invasion [8], and myeloid migration and infiltration [9].

How Fas signaling is able to trigger apoptotic vs. non-apoptotic cellular responses is an area of ongoing research. We and others have previously shown that pro-apoptotic signaling via Fas is highly sensitive to mutations that alter the structure of the receptor or downstream signaling complex. The extracellular preligand assembly domain (PLAD) [10,11], the transmembrane domain [12], and the localization of Fas to lipid raft microdomains [13,14,15] are all required for efficient apoptosis induction. Fas and FADD death domains assemble into an oligomeric structure defining a minimal signaling unit consisting of two trimers of Fas [1]. Similarly, studies with different versions of recombinant FasL defined a dimer of trimers as the minimal oligomer able to trigger apoptosis [16]. Recruitment of caspase-8 by FADD via the death effector domains and oligomerization of the death effector domains of caspase-8 activate the proteolytic activity of caspase-8, liberating the active form of caspase-8 into the cytoplasm and initiating the caspase cascade that results in apoptosis [17,18,19,20,21].

Most previous studies of Fas oligomerization at the plasma membrane have been limited by the use of engineered versions of soluble FasL or anti-Fas antibodies to trigger Fas signaling. Conversely, membrane-bound FasL is known to be the physiological trigger for Fas-induced apoptosis in vivo [22]. In addition, studies using confocal microscopy cannot provide sufficient resolution to determine the stoichiometry of Fas within signaling clusters. Here, we use structured illumination microscopy and the single-molecule-based technique of photoactivated localization microscopy (PALM) to investigate the behavior of Fas/CD95 on the plasma membrane. Using PALM imaging, we describe the clustering of Fas receptors at single-molecule resolution in response to FasL embedded in a lipid bilayer, simulating the interactions that occur when membrane-bound FasL interacts with cells expressing Fas. Using fluorescence correlation imaging, we defined the requirements for recruiting FADD to the plasma membrane, and we correlated these results with measurements of Fas–Fas and Fas–FADD interactions by flow-cytometric FRET analysis and apoptosis induction. Together, these results allow us to propose a model of Fas interacting with FasL and FADD in the plasma membrane at the nanoscale level.

## 2. Materials and Methods

### 2.1. Cell Culture and Transfection

Rapo C2 (Fas-deficient, kind gift from Marcus Peter), I2.1 (FADD-deficient) [23], and I9.2 (caspase-8-deficient) [24] Jurkat T cells were maintained at 37C in phenol-red free RPMI (Thermo Fisher Scientific 11835030, Waltham, MA, USA) supplemented with 10% FBS (company and cat), L-glutamine (Thermo Fisher Scientific 35050061), and penicillin-streptomycin (Thermo Fisher Scientific 1514048). For transfection, cells were concentrated to 1 × 10^7^ cells/mL, and 420 µL of cell suspension mixed with 15–20 µg of plasmid DNA was electroporated using a BTX ECM 630, as previously described [25]. All Fas fusion plasmids were constructed on a modified pEGFP-N1 vector (Clontech/Takara Bio, San Jose, CA, USA) with a Kozak-TNFR2 leader-HA tag sequence, flanked by NotI and SalI, introduced immediately upstream of the XhoI cut site. FADD death domain fusions were constructed on a pEYFP-C1 vector (Clontech/Takara Bio, San Jose, CA, USA). Caspase fusions were constructed on a pEGFP-N1 vector. All constructs were driven by the CMV promoter. The fluorescent proteins used for PALM imaging were PAGFP and PAmCherry. The mCerulean and mVenus fluorescent proteins were used for flow-cytometric FRET measurements.

### 2.2. Preparation of Supported Planar Lipid Bilayers

Liposomes and glass-supported planar lipid bilayers were prepared as previously described [26,27]. Briefly, liposomes were synthesized using a mixture of 1,2-dioleoyl-sn-glycero-3-phosphocholine (DOPC, Avanti Polar Lipids 850375, Alabaster, AL, USA) and either 1,2-dioleoyl-sn-glycero-3-phosphoethanolamine-N-(biotinyl) (Biocap PE, Avanti Polar Lipids 870282, Alabaster, AL, USA) at a 2% molar ratio to DOPC or 1,2-dioleoyl-sn-glycero-3-[(N-(5-amino-1-carboxypentyl)iminodiacetic acid)succinyl] (Ni NTA DGS, Avanti Polar Lipids 790404, Alabaster, AL, USA) at a 6% molar ratio to DOPC. Lipids were mixed in chloroform, lyophilized overnight, reconstituted in argon-purged, tris-buffered saline, sonicated, and extruded through 100 nm pore membranes (Avanti Polar Lipids 610005, Alabaster, AL, USA) using a mini-extruder kit (Avanti Polar Lipids 610000). Bilayers were formed in flow chambers assembled using coverslips cleaned with acid piranha solution (70% *v*/*v* sulfuric acid (Sigma Aldrich, St. Louis, MO, USA) and 30% vol./vol. hydrogen peroxide (Thermo Fisher Scientific). Liposomes were deposited onto clean coverslips and chambers were flushed with HBS buffer to form bilayers. Subsequent bilayers were washed with 5% casein solution (Sigma Aldrich C4765) to block nonspecific binding sites. Next, bilayers were incubated with monobiotinylated anti-CD3, His-tagged ICAM1 (Sino Biological 10346H08H, Beijing, China), streptavidin (Thermo Fisher Scientific SNN1001, Waltham, MA, USA) and Strep-tag Fas ligand. OKT3 clone anti-CD3 (Thermo Fisher Scientific 16003781, Waltham, MA, USA) was biotinylated using a commercial kit (Thermo Fisher Scientific 21925), adjusted to reduce the degree of labeling to ~1 biotin per antibody. Additionally, for SIM experiments, monobiotinylated anti-CD3 was labeled with Alexa Fluor 647 using a commercial kit (Thermo Fisher Scientific A20186, Waltham, MA, USA) as directed. The Fas ligand Strep-tag was synthesized in plasmid form, and harvested from transfected HEK 293T cells.

### 2.3. PALM Imaging and Clustering Analysis

PALM experiments were carried out by transient expression of fusion constructs of proteins of interest linked with either PAGFP [28] for single-color PALM, or PAGFP and either PAmCherry or PATagRFP for dual-color PALM [29]. Jurkat T cells were transfected ~24 h before being concentrated 10× and flushed into bilayer flow chambers. Preliminary observations showed that 10–15 min was the minimum time required to reproducibly observe spread out cells with enough surface area in the TIRF-zone for imaging, and it took 30–35 min before any clustering/heterogeneity was observed by diffraction-limited imaging. For these reasons, we chose 35 min as the standard incubation period unless otherwise noted, after which cells were fixed with 4% paraformaldehyde (Electron Microscopy Sciences 30525, Hatfield, PA, USA) and 0.1% glutaraldehyde (Electron Microscopy Sciences 16020, Hatfield, PA, USA) in PBS for ~30 min, quenched with 50 mm glycine (Sigma Aldrich 50046, St. Louis, MO, USA) in PBS for ~30 min, and labeled with 1:2000 diluted Tetraspek beads (T7279 Invitrogen, Waltham, MA, USA) as fiducial markers. Samples were imaged for a maximum of 3 h after fixation on a Zeiss Elyra PS.1 PALM microscope with a 100 × 1.46 NA objective using Zeiss Zen software (Carl Zeiss, Thornwood, NY, USA). For single-color PALM, PAGFP was activated and excited with a 488 nm laser, with additional activation by a 405 nm laser as necessary. For dual-color PALM, PAGFP was activated and excited with a 488 nm laser first, until either the image file was full or the cell was depleted of activatable PAGFP. Subsequently, PAmCherry or PATagRFP was activated with a 405 nm laser and excited with a 543 nm laser until either the image file was full or the cell was depleted of activatable PAmCherry. Laser intensities were adjusted during imaging to maintain a sparse population of activated, excited molecules in each image frame to maximize localization precision. Unless noted, 20,000 frames were collected at a 50 ms exposure time per frame to form a composite PALM image.

We used a modified Hoshen–Kopelman method-based cluster analysis [30] to obtain a quantitative description of the spatial organization of Fas receptors (Fas). The cluster analysis algorithm used an iterative spatial clustering search to identify individual clusters of Fas molecules and spatially isolated single molecules. We defined protein islands as spatially isolated Fas clusters. The calculated spatial coordinates and localization precision of the single molecules were used as metrics for assigning individual molecules to specific protein islands. We employed an iterative grouping process to identify all neighboring single molecules within a distance of 60 nm (∼3*σ* of the PSF of the imaging system), which were then assigned to the same protein island. The cluster analysis operation was performed over the composite super-resolution image, yielding the spatial location of individual protein islands and the distribution of proteins within the islands across the entire image. Following the identification of protein islands and constituent Fas molecules, various cluster parameters such as size, shape, density, and protein numbers of each individual protein island were computed. The size and shape analyses of protein islands were performed on islands containing at least three protein peaks, since at least three points are required to define a two-dimensional space. To define the space occupied by each protein island, we calculated a convex hull (the smallest convex set) for the set of molecules comprising each protein island, with the limiting lines joining the vertices of the convex hull serving to demarcate the protein islands. The areas of the protein islands were estimated from the area of the corresponding convex hulls. The protein island radius was considered to be equal to the radius of a circle of the same area as the convex hull of the protein island. The cluster density was calculated as the density of molecules within the convex hull. Statistical distributions of the different cluster parameters were obtained by combining the cluster parameter values of all the protein islands. This provided an accurate quantitative description of the heterogeneous Fas organization in the super-resolution composite images.

### 2.4. SIM Imaging

SIM experiments were carried out by transient expression of Fas-Emerald GFP, FADD (death domain)-mCherry, and caspase-8 (C360S)-TagRFP fusion constructs. Jurkat T cells were transfected ~24 h before being concentrated 10X and flushed into bilayer flow chambers. Live cells were imaged immediately after loading into flow chambers on a DeltaVision OMX SR microscope (GE Healthcare Life Sciences, Pittsburgh, PA, USA) with a 60 × 1.42 NA objective and standard 405-488-568-640 nm laser lines. Image processing was performed using GE SoftWorx software (GE Healthcare Life Sciences, Pittsburgh, PA, USA)

### 2.5. FasL-LZ Generation

FasL-LZ was generated through overexpression of the extracellular domain of Fas ligand fused to a FLAG-tag and an isoleucine zipper motif for receptor self-oligomerization. Transfected HEK 293T cell supernatants were collected and the protein was purified over magnetic beads conjugated to an anti-FLAG tag antibody (anti-FLAG M2 beads; Sigma, St. Louis, MO, USA), with quantitation performed by ELISA (R&D Systems, Minneapolis, MN, USA), as shown in Appendix A.

### 2.6. Fluorescence Resonance Energy Transfer

Full-length human Fas lacking the death domain (Fas ΔDD), or lacking the preligand assembly domain (ΔPLAD), were cloned into pEYFP-N1 (Clontech/TaKaRa Bio), with the C199V palmitoylation mutation introduced via site-directed mutagenesis. The plasmids encoding ALPS mutants A257D and E261K were kind gifts from Francis Chan. Human FADD death domain (FADD DD) was cloned into pECFP-N1 (Clontech/TaKaRa Bio). To obtain mCerulean3- and mVenus-tagged constructs, each vector had the respective fluorescent protein removed via restriction digestion (AgeI and BsrGI) and replaced with either mCerulean3 or mVenus. Each construct was transfected into Jurkat cells via electroporation as described above. Cells were analyzed 24 h post-transfection by flow cytometric detection of FRET using an LSR II Fortessa flow cytometer (Becton Dickinson, Franklin Lakes, NJ, USA) with 447/488 nm dual-laser excitation as previously described [31]. FRET data are shown for cells gated to comparable expressions of the Cerulean3 and Venus fusion proteins.

## 3. Result

### 3.1. Reconstitution of Fas Signaling Using FasL Embedded in a Supported Lipid Bilayer

In physiological settings, both the Fas receptor (Fas) and Fas ligand (FasL) are primarily present as proteins anchored in the plasma membrane (PM). Consequently, Fas–FasL interactions frequently occur at contact sites between adjoining cells. To correctly recapitulate the Fas–FasL interactions at membrane–membrane interfaces, we established a supported lipid bilayer (SLB) system displaying FasL for interaction with Fas on the PM of interacting lymphocytes (Figure 1A). The bilayer platform consisted of a fluid SLB composed of DOPC and trace amounts of biotinylated-DOPE and Ni-NTA lipids. Functionalization of the SLB with FasL was achieved by conjugating a Strep-tag-labeled FasL to biotinylated-DOPE via a streptavidin bridge. His-tagged ICAM-1 and anti-CD3 antibodies were also incorporated in the SLB via conjugation to Ni-NTA and biotinylated-DOPE, respectively, to facilitate the engagement of the cells with the SLB platform (Figure 1A). Jurkat T cells transfected with constructs encoding Fas molecules fused to a C-terminal fluorescent protein were placed in chambers containing SLBs and imaged by diffraction-limited TIRF microscopy. In control experiments, where FasL was absent from the SLB, Fas on the surface of T cells was distributed diffusely and showed no correlation with T cell receptor (TCR) clusters, highlighted by anti-CD3 antibodies (Figure 1B). When cells were placed on the SLB containing FasL to mimic the presentation of membrane-bound FasL to Fas on target cells, cell-surface Fas molecules reorganized into punctate structures. However, the Fas clusters were not clearly resolvable in the diffraction-limited TIRF images (Figure 1B). In addition, even after the engagement of Fas with SLB-bound FasL, Fas showed little colocalization with TCR clusters, indicating that Fas–FasL clusters were independent of the TCR under these conditions.

To more precisely measure the Fas reorganization following its engagement with SLB-anchored FasL, we used single-molecule-based photoactivated localization microscopy (PALM) to evaluate the nanoscale spatial distribution of Fas molecules fused to photoactivatable GFP (PA-GFP) [32] before and after engagement with FasL. Reconstructed PALM images of receptor probability density revealed that before stimulation, Fas is primarily distributed as small clusters across the PM (Figure 1C, left panel). However, following engagement with SLB-anchored FasL, the majority of Fas is reorganized into large clusters of high receptor density (Figure 1C, right panel). PALM images contain precise positional information for all detected Fas molecules organized into a complex pattern of molecular positions. We used a combination of spatial grouping algorithms and a modified Hoshen–Kopelman-based cluster analysis (HK analysis) to identify and visualize clusters of Fas molecules and spatially isolated individual Fas (i.e., monomers) in the complex super-resolution images. We refer to every individual spatially isolated protein assembly identified by the cluster analysis as a protein island. A spatially isolated single Fas molecule, by this definition, represents the smallest protein island (in terms of protein occupancy). The HK cluster analysis allowed us to identify individual protein islands and measure physical characteristics of islands, such as the size of island (r_island_, radius in nm), occupancy of island (ϕ_island_, number of proteins in a cluster), and density of island (ρ_island_). We used these parameters as quantitative descriptions of the spatial distribution of Fas. This allowed us to evaluate the differences in Fas spatial distribution under different conditions of stimulation or structural perturbations.

The distributions of Fas molecular positions color-coded according to occupancy of their assigned protein islands revealed that the stoichiometry of Fas in the plasma membrane changes significantly following stimulation by FasL (Figure 1D). Before FasL engagement, Fas molecules are present primarily as monomers (~40% of Fas) or small protein islands comprising 2–10 Fas molecules (−FasL, Figure 1D–F). Following interaction with FasL, receptor molecules redistribute into protein islands of higher occupancy, with the majority of Fas (~80%) present in islands containing >20 Fas molecules (+FasL, Figure 1D–F), which we refer to as “superclusters”. The highest occupancy protein islands following stimulation comprised up to 400–500 Fas molecules. Consistent with the transition of Fas molecules into islands of larger occupancy following Fas–FasL interaction, the percentage of protein island superclusters with >20 Fas molecules increased significantly whereas the percentage of low occupancy islands decreased (Figure 1G). In parallel with the change in occupancy, the size of the protein islands also increased after stimulation. Specifically, most of the Fas protein islands were smaller than 60 nm in radius before stimulation, whereas the majority of Fas protein islands exhibited radii >100 nm following FasL engagement (Figure 1H). Taken together, these results indicate that the interaction of Fas with membrane-bound FasL drives the reorganization of Fas receptors into protein superclusters at spatial scales significantly larger than the minimum hexameric Fas–FasL interaction units described in previous studies [1].

### 3.2. Fas Reorganization Requires Presentation of FasL on a Fluid Bilayer

In the reconstituted system used here to mimic the cell–cell interface, FasL is anchored to a fluid SLB comprising lipids with unsaturated acyl chains (DOPC and DOPE). The diffusive behavior of FasL is determined by the fluidity of the underlying lipid bilayer, with FasL in DOPC-SLB able to diffuse freely as in a mammalian plasma membrane. We examined whether membrane fluidity and the resulting diffusivity of FasL modulates the reorganization of the Fas molecules in the plasma membrane. To alter the fluidity of the membrane, we generated SLBs with saturated 1,2-distearoylsn-glycero-3-phosphocholine (DSPC) as the principal lipid component. DSPC has saturated acyl chains, and membranes composed of DSPC are in a gel phase at the physiological temperature, leading to reduced lateral mobility of embedded proteins [33]. Before stimulation with FasL, the distribution of Fas molecules on the PM of cells placed on a DSPC-SLB was similar to that on a DOPC-SLB, with Fas molecules primarily present as monomers or within small protein islands (r_island_ < 60 nm) of 2–10 Fas molecules (Figure 2A, left panel). However, engagement with FasL on a DSPC-SLB, in contrast to on a DOPC-SLB, did not change the spatial distribution of Fas. Receptor molecules persisted as a mixture of monomers and small islands of a few Fas molecules (ϕ_island_ < 10) (Figure 2A, left panel). Consistent with the lack of Fas clustering, the size of Fas islands on the DSPC-SLB did not increase after stimulation with FasL (Figure 2A, right panel).

As an alternative strategy for altering the diffusive behavior of FasL, we immobilized FasL, ICAM-1, and anti-CD3 molecules on polylysine-coated glass coverslips. This provided a platform for engaging Fas with completely immobile FasL. When cells were placed on the functionalized coverslips, Fas showed no change in spatial organization upon binding to FasL (Figure 2A). Fas was present primarily as small islands of monomers or a few Fas molecules both with and without stimulation with FasL. Taken together, these results indicate that the free diffusion of FasL and the fluidity of the underlying membrane are critical for the spatial reorganization of Fas molecules into larger protein islands.

### 3.3. Structural Requirements for Fas Reorganization into Nanoscale Superclusters

These results demonstrate that following engagement with FasL, Fas undergoes clustering at spatial scales significantly larger than would be predicted from the trimer–trimer Fas–FasL interface described in previous studies. This raises the question of whether the large proteins islands generated following Fas–FasL interactions represent functional signaling platforms. In such a scenario, the large-scale reorganization of Fas molecules presumably represents the first detectable physical event of Fas signaling. Earlier studies identified specific structural features of Fas that are critical for ligand-induced signaling, including receptor preassociation through the N-terminal preligand assembly domain (PLAD) [11], association with lipid raft microdomains [13,14,15], and a death domain competent at interacting with FADD [25]. To investigate the roles played by each of these features, we generated mutant versions of Fas and investigated their reorganization in the PM following the interaction of cells expressing these mutant receptors with SLB-anchored FasL. The FasR-C199V mutant is deficient in palmitoylation, the Fas-ΔDD mutant lacks the intracellular death domain, and the Fas-ΔPLAD mutant is missing the extracellular PLAD domain. Before ligand engagement, the three mutant Fas molecules exhibited spatial distributions that are similar to WT Fas, with the receptor molecules primarily present as monomers or small islands of <10 molecules (Figure 2C, left panel). Consistent with an impaired ability to form preligand oligomers [11], a slightly larger percentage of Fas-ΔPLAD was present as monomers, as compared to the other mutants and the WT receptor. Following interaction with FasL on the SLB, the mutant receptors exhibited variable extents of reorganization into larger protein islands (Figure 2C, left panel). For all three mutants, a smaller percentage of receptors were present in islands with high receptor occupancy (ϕ_island_ > 20) compared to WT Fas. Consistent with this, a smaller percentage of large protein islands (r_island_ > 100 nm) were generated following stimulation of the mutant receptors (Figure 2C, right panel). The Fas-C199V mutant was the least defective in forming large protein islands, followed by the Fas-ΔDD mutant, and then the Fas-ΔPLAD mutant, which was the most defective. These results indicate that all three mutants contribute to the transition of Fas molecules into large protein islands following interactions with FasL. Additionally, since all three mutants have previously been shown to be defective in Fas-mediated signaling, these results suggest that the large protein islands generated by Fas–FasL interaction in our model system likely represent an early signaling platform and are connected to downstream signal propagation.

To examine the contributions of structural motifs within the Fas DD to the receptor reorganization into large protein islands, we tested the behavior of two ALPS-associated Fas DD mutations that are located at a key interface for Fas–Fas and Fas–FADD death domain interactions, and that have been shown to severely disrupt and dominantly interfere with FADD recruitment [1,25]. To determine if the impaired signaling exhibited by these ALPS mutants is related to their ability to form large protein islands, we examined their spatial redistribution following the SLB-bound FasL engagement of two ALPS-associated Fas mutants, Fas-E261K and Fas-A257D. Strikingly, these two Fas mutants were even more defective in forming large protein islands than the Fas lacking the entire death domain. FasL engagement did not induce any significant reorganization of these ALPS-associated Fas mutants into larger protein islands either (Figure 2D). Instead, the mutants remained organized as small islands composed of either monomers or a few receptors (ϕ_island_ < 10 and r_island_ < 60 nm). These results indicate that the impaired signaling of ALPS mutants is likely related to their inability to form large protein islands following stimulation with FasL, and highlight the need for structural integrity of the Fas death domain and residues forming Fas–Fas and Fas–FADD interfaces to allow the formation of larger protein oligomers.

We previously used fluorescence resonance energy transfer between CFP- and YFP-tagged versions of Fas and the FADD-DD expressed in HEK 293T cells to study interactions between Fas receptors and Fas–FADD interactions [1,11,31]. These studies showed that baseline Fas–Fas interactions depend on the preligand association domain, and that FADD recruitment depends on the structural integrity of the Fas death domain. Here we refined this technique by expressing these receptors tagged with mCerulean3 and mVenus GFP variants, which have reduced spectral overlap, in the Fas-deficient Jurkat human T cell line RapoC2. Under these conditions, we were able to observe increases in FRET between CFP- and YFP-tagged Fas receptors in live cells after application of an oligomerized version of FasL stabilized by a leucine zipper (FasL-LZ), beginning at 5 min and stabilizing after 30 min (Figure 2E). The kinetics of this increase were similar to the formation of the large receptor protein islands seen using PALM. Comparing the increase in FRET between WT and the same Fas mutants we imaged using PALM, we found that both C199V-mutant Fas and ΔDD-mutant Fas display similar increases in FRET between CFP- and YFP-tagged receptors as WT Fas. The A257K and E261K Fas mutants, on the other hand, displayed a smaller, but still detectable, increase in FRET after addition of FasL-LZ. The ΔPLAD mutant, which is known to be defective in binding to FasL, exhibited no increase in FRET. To determine whether the FasL-driven increases in Fas proximity depend on the presence of FADD or caspase-8, we used mutant Jurkat cell lines lacking these key signaling proteins [23,24]. Strikingly, FasL-LZ induced similar increases in FRET in both FADD- and caspase-8-deficient Jurkat cells as in WT Jurkat cells (Figure 2F), showing that this increase in receptor proximity does not require FADD or caspase-8. A similar increase in the proximity of Fas-FP fusion proteins was observed in WT Fas-deficient Jurkat cells, indicating that endogenous Fas does not significantly affect the behavior of the fluorescent fusion proteins.

### 3.4. FADD Recruitment Following Interaction of Fas with SLB-Anchored FasL

The correlation between the impaired signaling of the different mutant Fas receptors and their impaired ability to form large protein islands supported the idea that the transition of Fas superclustering is a critical early event for signaling. To provide additional support for this idea, we investigated whether the protein islands formed by Fas–FasL interactions in our reconstituted system are capable of recruiting FADD, a critical next step in Fas signaling. To interrogate FADD recruitment and its association with Fas, we performed multicolor PALM to generate super-resolution images of Fas and FADD on the PMs of cells coexpressing Fas-PAGFP and FADD-PAmCh. PALM images were acquired following the interaction of cells with SLBs with or without FasL. We used pair cross-correlation analysis of the multicolor PALM images to obtain a quantitative measure of the nanoscale spatial association of Fas and FADD. Pair cross-correlation analysis provides an objective evaluation of the relative spatial organization of the two proteins, is insensitive to their relative concentrations on the PM, and can efficiently detect spatial association or segregation of molecules in single-molecule datasets with low spatial sampling. The pair cross-correlation function, *c*(*r*), generated by the analysis, has values >1 when there is spatial association between the two different molecules, has values <1 for spatial segregation of the molecules, and has a value equal to 1 when the molecules are randomly distributed with respect to each other.

Plots of pair cross-correlation functions revealed no significant association between Fas and FADD molecules before stimulation (Figure 3A, black curve). Strikingly, there is a significant increase in the association of FADD with Fas following Fas–FasL interaction, with the cross-correlation function showing positive spatial correlation (*c*(*r*) > 1) (Figure 3A, red curve) that declines with distance, becoming negligible at distances >100 nm between Fas and FADD. The strong association of FADD with Fas following Fas–FasL interactions in our reconstituted system raised the possibility that the formation of protein islands under these conditions is required for the recruitment of FADD to stimulated Fas receptors. To address this question, we examined the association of FADD with the different mutant versions of Fas that showed a diminished ability to form large protein islands following FasL engagement. We examined the spatial cross-correlation of FADD with three of the mutant receptors, Fas-C199V, Fas-ΔDD, and Fas-A257D, which exhibited different degrees of impairment in their ability to form large protein islands. All three of these mutant receptors showed no spatial correlation with Fas before stimulation (Figure 3B–D, black curves). Strikingly, following Fas–FasL interaction, only Fas-C199V exhibited increased association of FADD with Fas, though with reduced amplitude compared to WT Fas. The Fas-ΔDD and Fas-A257D mutants, on the other hand, showed no association with FADD even after ligand engagement (Figure 3B–D, red curves). Taken together, these results suggest that the transition of Fas into larger protein islands is critical for its ability to recruit FADD and to trigger the next steps in signal transduction. The increased association of FADD with Fas-C199V, albeit to a lesser extent than that with WT, is consistent with the ability of Fas-C199V to reorganize to some extent into large protein islands following interaction with FasL. The complete absence of FADD spatial association with the Fas-ΔDD mutant likely arises from the lack of DD domain in this mutant. For Fas-A257D, the lack of FADD association is consistent with its inability to form large protein islands.

Using flow-cytometric-based FRET between Fas and the FADD DD, we assessed the ability of WT and mutant Fas molecules to recruit FADD after application of FasL-LZ (Figure 3E). Compared to WT Fas, only the Fas-C199V palmitoylation mutant was able to recruit FADD, whereas the Fas-ΔDD, A257D and E261K mutants showed no increase in FRET after the addition of FasL. These data are consistent with the cross-correlation analysis of Fas and FADD using PALM imaging, and support the essential role of the death domain in recruitment of FADD, now measured in living cells.

### 3.5. Apoptosis Induction by Fas/CD95 Correlates with Ability to Form Large Protein Islands

To determine the functional significance of Fas receptor reorganization at the plasma membrane, we assessed the ability of WT Fas and selected Fas mutants to reconstitute apoptosis in Fas-deficient Jurkat cells after addition of FasL-LZ. Whereas the Fas-ΔDD and ΔPLAD mutants were completely incapable of inducing apoptosis, the Fas-C199V palmitoylation mutant induced apoptosis with an efficiency comparable to WT Fas (Figure 4A). As the efficiency of Fas-induced cell death correlates with Fas surface levels, it is possible that the relatively high levels of surface receptors in transfected Jurkat cells may have obscured a subtle cell-death defect in the C199V mutant, which was only partially defective in the formation of larger receptor oligomers after interaction with the SLB FasL. To overcome this problem, we used T cells from a mouse harboring a transgene encoding the murine-equivalent palmitoylation mutant back-crossed to the Fas-deficient lpr/lpr genotype (FasC194VTg ^lpr/lpr^) [31], and a newly-generated knock-in mouse line where the Fas-C194V mutant replaces the wild-type gene (FasC194V^KI/KI^, Appendix A). Like FasC194VTg ^lpr/lpr^ mice, FasC194V^KI/KI^ mice do not display lymphadenopathy, splenomegaly, accumulation of CD4^−^CD8^−^ peripheral T cells, or autoantibody formation, all of which are hallmarks of Fas deficiency (Appendix A). Levels of Fas on the surface of FasC194V^KI/KI^-activated T cells were similar to WT mice (Figure 4B), in contrast to the reduced surface expression of Fas seen in FasC194VTg^lpr/lpr^ mice [31]. Nevertheless, a partial defect in apoptosis induced by FasL-LZ in T cells from FasC194V^KI/KI^ mice could be observed as compared to WT T cells. Apoptosis in T cells from FasC194VTg^lpr/lpr^ mice, on the other hand, was almost as defective as in T cells from Fas deficient *lpr/lpr* mice (Figure 4C). Interestingly, when activated memory T cells were restimulated through their TCR, which leads to restimulation-induced cell death dependent on membrane-bound FasL [22], the defect in apoptosis was similar between FasC194VTg ^lpr/lpr^ and FasC194V^KI/KI^ mice (Figure 4D). This may be reflective of the greater efficiency of mFasL in inducing cell death than soluble FasL-LZ, and supports the physiological significance of palmitoylation and lipid raft microdomain association of Fas in promoting apoptosis, though apparently not required to prevent autoimmunity.

## 4. Discussion

Using super-resolution microscopy, we have defined the stoichiometry of Fas/CD95 before and after engagement with its physiological ligand, the membrane-bound FasL. Prior to ligand engagement, Fas is present on the plasma membrane at low levels of oligomerization, with 60% of receptors present as either monomers or dimers. This changes dramatically after addition of FasL, with more than 80% of the receptors now present in oligomers of more than 20 molecules and in clusters with a radius exceeding 100 nm. The positive correlation between the size of the Fas oligomers formed by the Fas mutants and their efficiency at inducing apoptosis supports the idea that formation of large receptor oligomers is crucial for Fas-mediated apoptosis. Non-apoptotic signaling may not require as great a degree of receptor oligomerization, as non-apoptotic functions of Fas, such as activation of AKT signaling and precocious T cell differentiation, are intact in FasC194VTg ^lpr/lpr^ mice [31], despite the reduced oligomerization seen with the equivalent palmitoylation mutant in human Fas. These results are consistent with the surface receptor clusters that have been visualized using confocal microscopy after stimulation with soluble [34] or membrane-bound [35] FasL, but allow a more quantitative understanding of the degree of receptor oligomerization. Although the minimal unit of receptor oligomerization may be a dimer of trimers, as suggested by the Fas–FADD DD crystal structure [1], the vast majority of receptors are in oligomers of more than 20 receptors after engagement of membrane-bound FasL. These oligomers are larger than those seen with PALM imaging of TNFR1 after engagement with soluble TNF, where the largest oligomers observed contained only nine receptors [36]. This may be due to the use of soluble rather than membrane-bound TNF as the ligand in that study, or to differences between TNFR1 and Fas signaling.

There are a number of limitations of these results that should be taken into account. Our imaging studies were carried out using transfected Fas-FP fusion proteins, which may be expressed at higher than physiological levels. Although we used versions of fluorescent proteins engineered to minimize clustering through the FP moiety, supraphysiological levels of Fas-FP may favor receptor clustering and may minimize defects in signaling for mutant receptors that exhibit only marginal defects in clustering. This may be the case for the Fas-C199V mutant, which was only partially defective in clustering in Jurkat cells, and which did not exhibit a defect in apoptosis, whereas Fas-C199V expressed at physiological levels in knock-in mice was partially defective in inducing apoptosis. To form effective conjugates between Fas-expressing Jurkat cells and the SLB, we used TCR and integrin ligands embedded in the SLB. Notably, Fas did not cocluster with the TCR in the TCR-nucleated immune synapse that forms under these conditions, indicating that FasL–Fas interactions likely occur independently of the immune synapse. The use of planar supported lipid bilayers allowed more effective super-resolution imaging of Fas receptors at the plasma membrane, and although SLBs may not be as physiological as anti-gen-presenting cells, many of the observations originally made with Jurkat-SLB immune synapses have been confirmed in primary T cells [37]. Even considering these limitations, our use of SLBs to display receptor ligands may be advantageous for studying the membrane reorganization of other receptors that are not part of the formation of antigen-receptor-driven supramolecular clusters. These results complement other studies that have imaged Fas receptor reorganization following ligation with soluble ligands [34,36] and solution-based clustering assays [38]. Overall, our results support the concept of superclustering as an essential step in signaling by TNF-family receptors [39].

## Figures and Tables

**Figure 1 cells-11-01908-f001:**
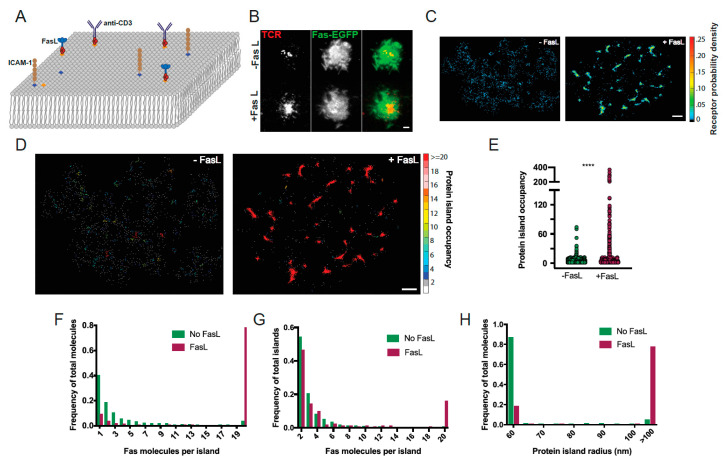
SIM and PALM imaging of Fas receptor clustering after interaction with SLB-anchored FasL. (**A**) Schematics of the experimental setup used to image the spatial distribution of Fas receptors following stimulation with Fas ligand (FasL). Jurkat cells expressing Fas-EGFP were adhered to supported lipid bilayers (SLBs) containing FasL, ICAM-1, and anti-TCR (anti-CD3) antibodies. (**B**–**D**) Spatial distribution of Fas receptors on plasma membrane of Jurkat cells placed on SLBs. (**B**) Diffraction-limited TIRF images showing the distribution of Fas-EGFP (green) T cell receptors (labeled with anti-CD3 antibodies, red), with (bottom panel) or without (top panel) FasL added to the SLB. Scale bar = 2 microns. (**C**) PALM images of nanoscale spatial distribution of Fas-PAGFP receptor fusion proteins before (right panel) or after (left panel) engagement with FasL. The relative probability density of Fas receptors is mapped on to each pixel using the density color map on the right. Scale bar = 1 micron. (**D**) Spatial distribution of protein islands identified by cluster analysis of Fas PALM images acquired before (left) or after (right) stimulation with FasL. Individual receptors within a protein island are color-coded according to the number of Fas receptors (protein island occupancy, ϕ_island_) present in the protein island. Scale bar = 1 micron. (**E**–**G**) Plot of physical parameters of the protein islands obtained from spatial cluster analysis of the distribution of Fas receptors in PALM images. (**E**) Plot of distribution of number of Fas receptors per protein island (ϕ_island_) with or without stimulation with Fas ligand. Each circle represents a single protein island. Data are from four cells for each condition and are compared by two-tailed Mann–Whitney *U* test (*p* < 0.0001). (**F**) Plot of distribution of total Fas receptors that reside in protein islands containing indicated number of receptors. Data are shown for both with (magenta) or without (green) stimulation with Fas ligand. (**G**) Plot showing the frequency of occurrence of protein islands containing indicated number of Fas receptors (ϕ_island_). (**H**) Plot of percentage of total Fas receptors residing in protein islands of indicated size (radius in nm).

**Figure 2 cells-11-01908-f002:**
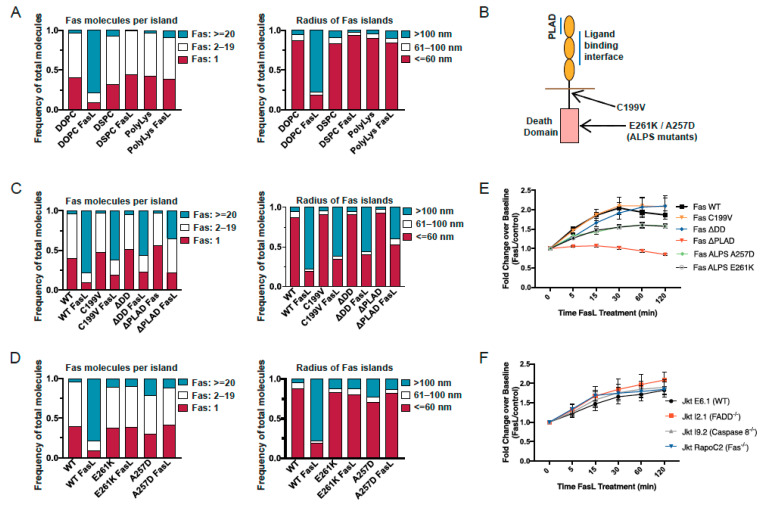
Contributions of structural domains of Fas to nanoscale receptor clustering. (**A**) Spatial distribution of Fas receptors was quantified following interaction of Jurkat cells with different FasL-containing substrates. FasL was either incorporated into fluid DOPC-based supported lipid bilayers, gel-like DSPC-based supported lipid bilayers, or immobilized onto polylysine-coated coverslips. Cluster parameters were evaluated with or without stimulation with FasL. (Left panel) Plot of distribution of percentage of Fas receptors in protein islands containing indicated number of proteins. (Right panel) Distribution of percentage of proteins residing in protein islands of indicated size (radius in nm). (**B**) Schematic of Fas receptor showing the location of preligand assembly domain (PLAD), the palmitoylation site CysC199, the death domain, and the ALPS mutations. (**C**,**D**) Cluster parameters of protein islands of wild-type Fas or indicated Fas receptor mutants (C: structural mutants, D: ALPS mutants) with or without interactions with FasL. Plots of distribution of percentage of proteins in protein islands of indicated protein occupancy (left panel, **C**,**D**), or within protein islands of indicated size (right panel, **C**,**D**). (**E**) Fas-deficient Jurkat cells (RapoC2) were cotransfected with homotypic pairs of indicated receptors, each tagged with either mCerulean3 or mVenus. 24 h post-transfection, cells were treated with FasL-LZ, and FRET between the pairs of Fas molecules was measured at the indicated timepoints. Fold change was calculated as the change in FRET intensity over similarly transfected, but untreated, cells. A minimum of *n* = 4 was performed for each experimental condition. (**F**) Wild-type Fas receptors tagged with either mCerulean3 or mVenus were cotransfected into wild-type Fas-sufficient Jurkat cells or mutant Jurkat cells deficient for Fas (RapoC2), FADD (I2.1), or caspase-8 (I9.2). Cells were stimulated with FasL-LZ, as in (**E**), and assessed for Fas–Fas FRET via flow cytometry on the Cerulean3^+^Venus^+^ viable population. Fold change was calculated as in (**E**). A minimum *n* = 3 was performed for each experimental condition.

**Figure 3 cells-11-01908-f003:**
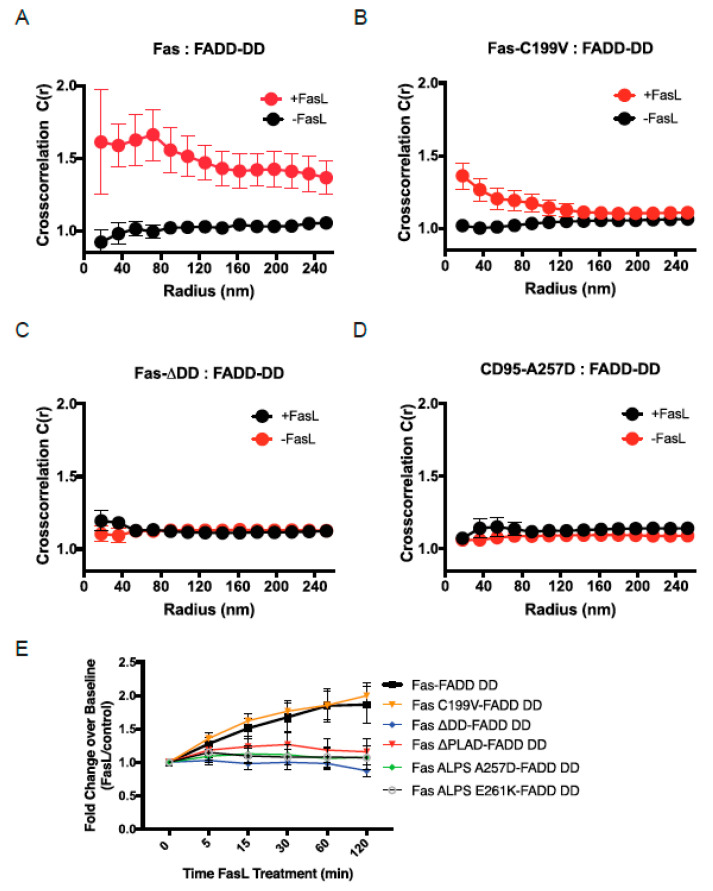
Cross-correlation and FRET analysis of Fas–FADD interactions. (**A**–**D**) Spatial interactions of FADD with either wild-type Fas or different mutants of Fas receptors on plasma membrane of Jurkat cells were quantified by spatial cross-correlation analysis of multicolor PALM images. PAGFP-tagged receptor molecules and FADD-PAmCh were coexpressed in Jurkat cells and imaged with or without interactions with FasL on DOPC-based fluid supported lipid bilayers. Plots of spatial cross-correlation (*c*(*r*)) between FADD-PAmCh and PAGFP-labeled wild-type Fas receptor (**A**), palmitoylation deficient Fas-C99V (**B**), Fas-ΔDD lacking death domain (**C**), or the ALPS mutant Fas-A257D (**D**). Value of *c*(*r*) >1 indicates increased spatial clustering of the two kinds of molecules, *c*(*r*) < 1 indicates spatial exclusion, and *c*(*r*) = 1 indicates random distribution of the molecules. (**E**) Fas-deficient RapoC2 Jurkat cells were cotransfected with a Cerulean3-tagged FADD containing only the death domain (FADD DD) and either wild-type or mutant Fas receptors as indicated. Cells were treated with FasL-LZ and changes in FRET between Fas and FADD were analyzed as in Figure 2E,F. A minimum of three independent experiments were performed (*n* = 3).

**Figure 4 cells-11-01908-f004:**
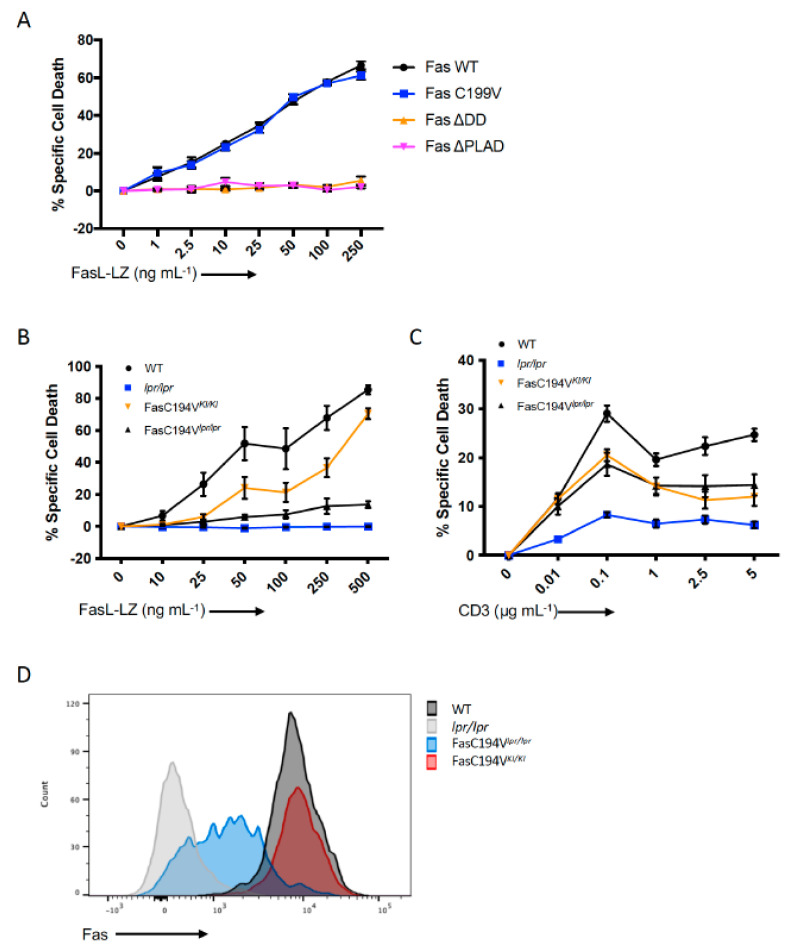
Relationship between Fas clustering and apoptosis induction in T cells. (**A**) Fas-deficient Jurkat cells (RapoC2) were transfected with mVenus-tagged WT or mutant Fas. 24 h post-transfection, cells were treated with increasing amounts of FasL-LZ for 6 h prior to staining with Annexin V, DiIC(5), and Live/Dead, and being analyzed by flow cytometry. (**B**) Activated CD4^+^ T cells from mice with the indicated genotype were stained for Fas surface expression and assessed by flow cytometry. Data are representative of *n* = 3 independent experiments. (**C**) CD4 T cells were isolated from mice of the indicated genotypes and activated for 72 h with anti-CD3/28. Cells were cultured with supplemental IL-2 prior to treatment with FasL-LZ at the indicated concentrations for 6 h, then stained for assessment of cell death via flow cytometry as in (**A**). (**D**) CD3/28 activated CD4 T cells from mice of the indicated genotypes were rested in IL-2 and subsequently restimulated with plate-bound anti-CD3 antibody at the indicated concentrations for 16 h prior to assessment for death via flow cytometry as in (**A**). Panels (**A**–**C**) are cumulative data from a minimum of three independent experiments (*n* = 3).

## Data Availability

Data are available from the authors according to the policies of the NIH intramural research program and HHMI.

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
