# Peer review of "Super-Resolution Imaging of Fas/CD95 Reorganization Induced by Membrane-Bound Fas Ligand Reveals Nanoscale Clustering Upstream of FADD Recruitment"

_cells, 2022, doi:10.3390/cells11121908_

Round 1
Reviewer 1 Report
The manuscript titled 'Super-resolution imaging of Fas/CD95 reorganization induced by membrane-bound Fas ligand reveals nanoscale clustering upstream of FADD recruitment' presents a compelling story identifying a previously misidentified stoichiometry for Fas receptors and further exploring the underlying mechanism regulating the observed changes in Fas receptor clustering. The manuscript is well written, presented with carefully planned experiments and compelling data verified by multiple different approaches to confirm the critical role of Fas receptor clustering and its different sub domains in induction of apoptosis.
Major concerns: 1) Does the presence of the GFP fusion protein on Fas alter receptor oligomerization? Could this contribute to the larger protein islands observed as the fp tag could potentially lead to increased oligomerization? How have the authors verified or negated the effect of FP tag on their observations and quantifications?
2) Following engagement of Fas with SLB bound FasL, it showed little colocalization with TCR clusters. What are the physiological implications of these findings?
3) How does the fluidity of the the synthesized DOPC-SLBs compare to cell membranes in cells? The DOPC-SLB is a minimalistic artificial bilayer which does not comprise all components of the T cell membrane. Could this potentially contribute to the observed differences in oligomerization states and protein island occupancies?
4) How do the oligomerization states evolve with time post Fas ligand stimulation? The experiments presented within the manuscript were performed ~ 35 minutes post exposure (time when cells were fixed)? How does this timeline compare to references cited within the manuscript (Ref 1) which shows distinct spatial scales for Fas receptor clusters?
Minor concerns: 1) The authors should provide more details for the electroporation method employed for transfection.
2) The data for protein purification and quantification performed by ELISA should be shown in supplemental information.
3) In the methods section under FRET, the authors claim that the FRET data shown are gated for comparable expression of Cerulean3 and Venus FPs. How was this performed?
Author Response
We thank the reviewers for their comments and have addressed them fully in the replies in italics below and with changes to the figures as requested. We have updated the figures with changes requested, and also updated figure 3 with increased smoothing of the cross-correlation data that was intended to be included in the original submission. A revised manuscript with changes in yellow highlighting is also attached
Point-by-point response to reviewer 1
The manuscript titled 'Super-resolution imaging of Fas/CD95 reorganization induced by membrane-bound Fas ligand reveals nanoscale clustering upstream of FADD recruitment' presents a compelling story identifying a previously misidentified stoichiometry for Fas receptors and further exploring the underlying mechanism regulating the observed changes in Fas receptor clustering. The manuscript is well written, presented with carefully planned experiments and compelling data verified by multiple different approaches to confirm the critical role of Fas receptor clustering and its different sub domains in induction of apoptosis.
Major concerns: 1) Does the presence of the GFP fusion protein on Fas alter receptor oligomerization? Could this contribute to the larger protein islands observed as the fp tag could potentially lead to increased oligomerization? How have the authors verified or negated the effect of FP tag on their observations and quantifications?
The PA-GFP used in the study is derived from monomeric versions of EGFP and has the A206K mutation which minimizes oligomerization. We observe a significant increase in the extent of Fas clustering and Fas cluster size following engagement of Fas-PAGFP with FasL, indicating that the redistribution of Fas into clusters is primarily driven by ligand engagemen. In the absence of FasL, Fas-PAGFP does not show similar clustering even when expressed at comparable densities. We also observed differences in clustering for the different Fas mutants, suggesting that the structural properties of Fas modulate the extent of clustering following ligand engagement. We have wording in the discussion to acknowledge that even the monomeric PAGFP/EGFP version could induce some clustering at high density, but our results indicate that it does not play a dominant role in our experimental conditions since we can still detect significant differences in clustering for different conditions and protein constructs. We also added the pertinent references for PAGFP and mCherry to the methods section 2.3, and expanded the caveats related to using FP-fusion proteins already in the discussion section.
2) Following engagement of Fas with SLB bound FasL, it showed little colocalization with TCR clusters. What are the physiological implications of these findings?
the lack of Fas at the IS suggests that Fas-FasL interactions likely occur outside of the immune synapse which is nucleated by the TCR. We have added wording to this effect in the discussion section.
3) How does the fluidity of the the synthesized DOPC-SLBs compare to cell membranes in cells? The DOPC-SLB is a minimalistic artificial bilayer which does not comprise all components of the T cell membrane. Could this potentially contribute to the observed differences in oligomerization states and protein island occupancies?
We acknowledge that DOPC-SLB is a minimal system used to aid super-resolution imaging of receptor behavior on T cells, many of the observations of originally made with Jurkat-SLB immune synapses have been confirmed in primary T cells. If FasL is presented as a transmembrane protein on the surface of a cell, since transmembrane proteins diffuse slower than lipids, there is a possibility that there would be differences in clustering if the dynamics of ligand plays a significant role in Fas receptor clustering. But high/super-resolution imaging at the interphase of two cells would be a challenging proposition and beyond the scope of this paper. We have added wording to this point in the discussion section
4) How do the oligomerization states evolve with time post Fas ligand stimulation? The experiments presented within the manuscript were performed ~ 35 minutes post exposure (time when cells were fixed)? How does this timeline compare to references cited within the manuscript (Ref 1) which shows distinct spatial scales for Fas receptor clusters?
The minimum time between adding cells to the coverslip, sealing the chamber, and mounting on the microscope was 10 minutes. We found that 10-15 min was the minimum time required to reproducibly observe spread-out cells with enough surface area in the TIRF-zone for imaging, and it took 30-35 minutes before any Fas clustering was observed by diffraction limited imaging. This was the reason for choosing 35 minutes as the standard observation time, so that we would likely see differences in spatial organization following ligand engagement. We have added wording in the methods section to explain this reasoning.
Minor concerns: 1) The authors should provide more details for the electroporation method employed for transfection.
We have added a reference to the methods section with more details on the electroporation method
2) The data for protein purification and quantification performed by ELISA should be shown in supplemental information.
We have added a supplemental figure (S1) with the details of purification and quantitation of FasL to the revised manuscript
3) In the methods section under FRET, the authors claim that the FRET data shown are gated for comparable expression of Cerulean3 and Venus FPs. How was this performed?
Cells were gated on the Cerulean+Venus+ population, selecting only the cells where the fluorescence of each fluorophore had linearly equivalent mean fluorescent intensity.
Reviewer 2 Report
Frazzette et al. provide a study on FasR clustering induced by FasL using SMLS. The authors do not only look at FasL-FasR interactions as such, but also at the involvement of FADD. Overall I believe that this is a nice study, which adds important information to further decipher TNFRSF signaling. However, a few points need to be addressed before publication can be recommended:
1) Several studies have shown that the average inter-ligand distance between FasL is an essential factor for efficient apoptosis induction. Unfortunately, the authors do not characterize their FasL/anti-CD3/ICAM-1 modified SLB. The authors should show what the average distance between FasL is (based on density as e.g. shown in https://doi.org/10.1016/j.celrep.2019.10.054).
2) There should be a more throrough description of how data were obtained/data analysis was obtained in the methods (i.e. the spatial grouping algorithm and rge modified HK analysis for cluster identification and visualization).
3) There are no scale bars in any of the microscopy images.
4) P5 L223: "...larger than the minimum pentameric-hexameric Fas-FasL interactions..." --> Neither in the cited reference [1] nor in other literature have I read anything about pentamers. Are the authors sure they cited the correct paper here?
5) Fig. 1 F and G: The axis labeling is very confusing. Maybe a schematic or other guide would help to understand the graphics better without having to study the caption in depth.
6) What was the distance between proteins that was used for the authors to class the ligands/receptors as NON monomeric?
7) Fig. 2a: Again, it would help to change the axis descriptions somewhat. It is completely unclear if DOPC FasL means that this is FasL alone or if this is the distribution of FasR after stimulation with FasL.
8) p9, line396 "...that declines with distance, becoming neglibible at distances..." it is not clear to me which distance is meant here: the distance between FasRs? FasLs? or FasR and FADD?
9) In the discussion p.12 l484, "...to ligand engagement, FasL is present..." Do the authors really mean FasL or FasR?
Author Response
We thank the reviewers for their comments and have addressed them fully in the replies in italics below and with changes to the figures as requested. We have updated the figures with changes requested, and also updated figure 3 with increased smoothing of the cross-correlation data that was intended to be included in the original submission. A revised manuscript with changes in yellow highlighting is also attached
Point-by-point response to reviewer 2
Frazzette et al. provide a study on FasR clustering induced by FasL using SMLS. The authors do not only look at FasL-FasR interactions as such, but also at the involvement of FADD. Overall I believe that this is a nice study, which adds important information to further decipher TNFRSF signaling. However, a few points need to be addressed before publication can be recommended:
- Several studies have shown that the average inter-ligand distance between FasL is an essential factor for efficient apoptosis induction. Unfortunately, the authors do not characterize their FasL/anti-CD3/ICAM-1 modified SLB. The authors should show what the average distance between FasL is (based on density as e.g. shown in https://doi.org/10.1016/j.celrep.2019.10.054).
We did not fluorescently tag the FasL molecules in our experimental system, so we are not able to provide an estimate of the distance between FasL molecules, The average spatial localization of our PALM imaging is ~20 nm, so we do not have the resolution to interrogate actual receptor/ligand distance, which is likely to be significantly less than 20nm. Because of the close proximity of Fas and FasL, the intermolecular distances between FasL molecules are likely to be similar to what we estimated for Fas.
- There should be a more throrough description of how data were obtained/data analysis was obtained in the methods (i.e. the spatial grouping algorithm and rge modified HK analysis for cluster identification and visualization).
We have added a paragraph to section 2.30 in the methods detailing the clustering analysis method
3) There are no scale bars in any of the microscopy images.
We apologize for the omission, and we have added scale bars in the images included in the revised submission.
4) P5 L223: "...larger than the minimum pentameric-hexameric Fas-FasL interactions..." --> Neither in the cited reference [1] nor in other literature have I read anything about pentamers. Are the authors sure they cited the correct paper here?
We thank the reviewer for noting this error,and have corrected this to hexameric (dimer of trimer) interactions, which was what was meant.
5) Fig. 1 F and G: The axis labeling is very confusing. Maybe a schematic or other guide would help to understand the graphics better without having to study the caption in depth.
We thank the reviewer for highlighting this issue. We have revised the x and y-axis labels for Figure 1 F&G to clarify that Panel F displays the fraction of total proteins present in islands with a certain number of proteins, while Panel G displays the fraction of islands with a certain number of proteins.represents the total occupancy of protein islands.
6) What was the distance between proteins that was used for the authors to class the ligands/receptors as NON monomeric?
We have explained the analysis strategy in the Methods section 2.3. Briefly, an isolated receptor with no other molecules present within a radius of 60 nm around it was considered to be a monomeric receptor.
7) Fig. 2a: Again, it would help to change the axis descriptions somewhat. It is completely unclear if DOPC FasL means that this is FasL alone or if this is the distribution of FasR after stimulation with FasL.
We changed the labels for figures 2 A,C and D to be more clear. We apologize for the confusion about whether the distribution of FasL or Fas is given. All our single molecule imaging was done with labeled FasR, FasL is unlabeled and is not visible. So, all the data pertains to distribution of Fas (i.e. Fas receptors) on the T-cell membrane. We have made that clear in the revised images. In this particular case, the data corresponds to distribution of Fas (Fas receptors) when they engage with FasL presented on DOPC SLBs
8) p9, line396 "...that declines with distance, becoming neglibible at distances..." it is not clear to me which distance is meant here: the distance between FasRs? FasLs? or FasR and FADD?
We thank the reviewer for pointing this out, and have added wording to clarify that this refers to distances between Fas and FADD.
9) In the discussion p.12 line 484, "...to ligand engagement, FasL is present..." Do the authors really mean FasL or FasR?
We thank the reviewer for catching this error and have changed to Fas receptors as intended.